# Loss of Mature Lamin A/C Triggers a Shift in Intracellular Metabolic Homeostasis via AMPKα Activation

**DOI:** 10.3390/cells11243988

**Published:** 2022-12-09

**Authors:** Ying Zhou, Jia-Jie Yang, Yuan Cheng, Ge-Xuan Feng, Rong-Hui Yang, Yuan Yuan, Li-Yong Wang, Miao Wang, Lu Kong

**Affiliations:** 1Department of Biochemistry and Molecular Biology, Capital Medical University, Beijing 100069, China; 2Department of Physiology, Capital Medical University, Beijing 100069, China; 3Department of Pathology, Capital Medical University, Beijing 100069, China; 4The Central Laboratory for Molecular Biology, Capital Medical University, Beijing 100069, China; 5Department of Pathology, Beijing Friendship Hospital, The Second Clinical Medical College of Capital Medical University, Beijing 100050, China

**Keywords:** *LMNA*, AMPK, lipid metabolism, hepatocellular carcinoma

## Abstract

The roles of lamin A/C in adipocyte differentiation and skeletal muscle lipid metabolism are associated with familial partial lipodystrophy of Dunnigan (FPLD). We confirmed that *LMNA* knockdown (KD) in mouse adipose-derived mesenchymal stem cells (AD-MSCs) prevented adipocyte maturation. Importantly, in in vitro experiments, we discovered a significant increase in phosphorylated lamin A/C levels at serine 22 or 392 sites (pLamin A/C-S22/392) accompanying increased lipid synthesis in a liver cell line (7701 cells) and two hepatocellular carcinoma (HCC) cell lines (HepG2 and MHCC97-H cells). Moreover, HCC cells did not survive after *LMNA* knockout (KO) or even KD. Evidently, the functions of lamin A/C differ between the liver and adipose tissue. To date, the mechanism of hepatocyte lipid metabolism mediated by nuclear lamin A/C remains unclear. Our in-depth study aimed to identify the molecular connection between lamin A/C and pLamin A/C, hepatic lipid metabolism and liver cancer. Gain- and loss-of-function experiments were performed to investigate functional changes and the related molecular pathways in 7701 cells. Adenosine 5’ monophosphate-activated protein kinase α (AMPKα) was activated when abnormalities in functional lamin A/C were observed following lamin A/C depletion or farnesyltransferase inhibitor (FTI) treatment. Active AMPKα directly phosphorylated acetyl-CoA-carboxylase 1 (ACC1) and subsequently inhibited lipid synthesis but induced glycolysis in both HCC cells and normal cells. According to the mass spectrometry analysis, lamin A/C potentially regulated AMPKα activation through its chaperone proteins, ATPase or ADP/ATP transporter 2. Lonafarnib (an FTI) combined with low-glucose conditions significantly decreased the proliferation of the two HCC cell lines more efficiently than lonafarnib alone by inhibiting glycolysis or the maturation of prelamin A.

## 1. Introduction

Lamins on the nuclear envelope are members of the type V intermediate filament family and exist as two types in vertebrates: A and B, which are encoded by *LMNA* and *LMNB1/B2*, respectively [1,2]. *LMNA* is located on chromosome 1q21.2 and contains twelve exons. In normal conditions, human *LMNA* is alternatively spliced into two major transcript variants [3]: isoform A (lamin A, 74 kDa) and isoform C (lamin C, 65 kDa) [4]. Compared to the lamin A transcript, the lamin C transcript lacks exons 11 and 12 but contains additional 123 bases of intron 10 following downstream of exon 10. Prelamin A is a precursor protein and is generated by functional lamin A through four enzymatic processing steps: farnesylation (which guides prelamin A insertion into the nuclear membrane) at a CaaX box (cysteine, two aliphatic amino acids, any amino acid), proteolysis of the last three amino acid residues at the carboxy terminus, carboxymethylation, and finally cleavage of the C-terminal 15 amino acids [5,6,7]. The first 566 amino acids for mature lamin A and C are identical. The difference in their C-terminus is some unique 80 amino acids on lamin A and six different amino acids on lamin C [3]. The polymerization of mature lamins A and C and B-type lamins occurs in the nuclear membrane, assembles the nuclear pore complex and maintains chromatin function, and is involved in many critical molecular events such as transcription, replication, mitosis, cell cycling, apoptosis, and even cell differentiation [8,9]. Phosphorylation of mature lamin A/C is an important post-translational modification (PTM) during cell mitosis. Lamin A/C may be depolymerized by phosphorylation at N-terminal Ser22 or the C-terminal Ser392, and thus pLamin A/C at these two sites is commonly used as an indicator of mitosis and cell proliferation [10,11].

To date, many reports on *LMNA*-related diseases have been published. For example, some evidence supports the function of *LMNA* as an oncogene in gastrointestinal tract tumors [12], prostate cancer (PC) [13], hepatocellular carcinoma (HCC) [14,15], and metastatic colorectal cancer (CRC) [16,17,18]. In addition, mutations in *LMNA* induce common laminopathies characterized by multiple rare clinical symptoms, including muscular dystrophy, lipodystrophy, diabetes, dermopathy, neuropathy, leukodystrophy, and progeria [19,20,21]. Childhood progeria syndrome (Hutchinson–Gilford syndrome, HGPS) is due to the toxic protein progerin, a truncated variant caused by a point mutation at codon 608 of prelamin A [22]. Several studies have also clarified the underlying mechanism of *LMNA* mutations in progeria-induced aging and have shown the key function of lamin A/C in promoting the differentiation of myocytes and adipocytes [23,24,25]. However, the relationship between lipid metabolism and lamin A/C regulation remains unknown. A case report noted that two biological sisters with *LMNA*-mutant FPLD developed hypopharyngeal squamous cell carcinoma [26]. This study inspired us to focus on the role of lamin A/C in lipid metabolism associated with metabolic syndrome and HCC.

The liver is a central organ that is well known for coordinating the metabolic balance in the human body, and more than 80% of de novo fatty acid synthesis occurs in the liver and adipose tissue. Notably, the characteristic of FPLD is lipoatrophy of the extremities and, on the contrary, visceral fat deposition, insulin resistance and hypertriglyceridemia [27]. Aberrant expression of *LMNA* without mutations has also been reported in humans with obesity and type 2 diabetes [28,29]. HCC development is closely related to metabolic syndromes such as fatty liver and insulin resistance [30]. Thus, metabolic syndrome and HCC should share lipogenesis and factors associated with lamin A/C regulation. Here, we investigated the mechanism by which mature lamin A/C modulates triglyceride (TG) synthesis.

Our data reveal that lamin C interacts with phosphorylated adenosine 5’ monophosphate-activated protein kinase α (AMPKα)-Threonine 172 (T172) in the nucleus and that lamin A/C and its partners positively regulate the nuclear ATP biosynthesis. Therefore, we established a link between lamin A/C and energy metabolism.

## 2. Materials and Methods

### 2.1. Cell Lines, LMNA-KO or LMNA-KD Plasmids and Transfection

The 7701, HepG2 and HEK293T cell lines were obtained from the Department of Cell Biology at Capital Medical University. The MHCC97-H cell line was obtained from the Beijing Institute of Hepatology at YouAn Hospital at Capital Medical University. All cell lines were authenticated. *LMNA*-KO 7701 cells were generated using CRISPR–Cas9 technology. *LMNA*-KD and AD-MSCs were generated using shRNA technology. Six *LMNA*-sgRNA primer pairs and three *LMNA*-shRNA primer pairs were designed from the human GeCKO v2 libraries and the Sigma website, respectively (listed in Appendix A). The vectors used in the experiment included pLenti Guide-puro-Vector (Addgene, #52963, Beijing, China), pCDH-Cas9-2A-GFP-BSD (GENEWIZ, Suzhou, China), PLKO.1-puro-vector (Addgene, #8453, Beijing, China), and pLKO.1 Puro shRNA scramble (Addgene, #162011, Beijing, China). The sgRNA pairs were individually cloned into the pLenti Guide-puro-Vector, and shRNA pairs were cloned into the PLKO.1-puro-vector. Two micrograms of plasmids were transfected into 5 × 10^5^ cells with Lipofectamine 3000 (Thermo Fisher Scientific, Waltham, MA, USA) for 6 h, after which the culture medium was changed. GFP-positive monoclonal cells were sorted using flow cytometry (FACS Aria II, BD, New Jersey, USA) and cultured in DMEM (8119438, Gibco, New York, USA) or F12 complete medium (SH30023.01, HyClone, Thermo Fisher Scientific) supplemented with puromycin. After one month, stable cells lacking *LMNA* or *LMNA* knockdown were acquired. Western blotting verified the *LMNA*-KO or LMNA-KD efficiency in the cells, and wild-type (WT) cells or scrambled shRNA-expressing cells were used as a control.

### 2.2. Construction of the LMNA cDNA and Pre–LMNA and Mutant Plasmids

The untagged human *LMNA* (lamin C) (NM_005572) clone (#SC321549) and the tagged human *LMNA* (lamin A) (NM_170707) Clone (#RC204970) were purchased from OriGene Technologies (Jiangsu, China). PCR products containing the *LMNA/C* cDNA or different mutations were purified, sequenced (Rui Bo Xing Ke Company, Beijing, China) and then cloned into the pCMV6-AC vector (PS100020, OriGene) or pCMV6-AC-GFP vector (PS100010, OriGene) to construct plasmids to express lamin A, lamin C and the different mutants. Two restriction endonucleases (BamHI and NotI) were used in the cloning process. The plasmids were extracted with the Endotoxin-Free Plasmid Medium Extraction Kit (CW2105s, CW Biotech, Jiangsu, China) and transfected into cells with Lipofectamine 3000 (Invitrogen, 2105082, Carlsbad, USA). Western blot analysis was used to verify expression. The primer sequences are listed in Supplemental Appendix A.

### 2.3. Immunohistochemistry (IHC)

Paraformaldehyde-fixed human liver and adipose tissues (Gift of Professor Wang Miao, Department of Anatomy, Capital Medical University) were subjected to IHC to detect lamin A or pLamin A. Normal liver tissues are para-cancerous normal liver tissues, and normal adipose tissues are para-lipoma normal tissues. The paraffin sections were dehydrated, followed by antigen retrieval and incubation with hydrogen peroxide. Then, primary antibodies against lamin A/C (ab133256, 1:200, Abcam, Cambridge, UK) and Lamin A + Lamin C (phospho S392, ab58528, Abcam, 1:2000) were incubated with the sections overnight at 4 °C. The next day, the sections were incubated with a secondary antibody (Horseradish Peroxidase, HRP-conjugated goat anti-mouse/rabbit IgG polymer, PV-6000, ZSGB-BIO, Shanghai) for 1 h. The chromogenic reaction was terminated with DAB substrate (ZLI-9017, ZSGB-BIO, Shanghai, China). The localization of antibody-positive areas was recorded using a light microscope (Olympus, BX41, Tokyo, Japan), and the sections were photographed.

### 2.4. RNA Extraction and qRT–PCR

Total RNA was extracted with TRIzol (Invitrogen, Carlsbad, CA, USA), FastKing gDNA Dispelling RT SuperMix (KR118-02, Tiangen, Beijing, China) and PowerUp^TM^ SYBR^TM^ Green Master Mix (A25742, Applied Biosystems, Carlsbad, CA, USA) according to the instructions accompanying TRIzol. All qRT–PCR primer sequences are listed in Supplemental Appendix A.

### 2.5. Detection of the Intracellular TG and Lactic Acid Contents

The intracellular TG concentration was measured with a kit according to the instructions (E1013, Applygen, Beijing, China). First, the cells were washed with PBS and then lysed at 25 °C for 10 min. Then, the protein in 3 μL of lysate was quantified with a BCA protein quantification kit (P0011, Beyotime, Shanghai, China). The remaining supernatant was transferred to a 1.5-mL centrifuge tube, heated at 70 °C for 10 min, and then centrifuged at 2000 rpm for 5 min at 25 °C, after which the supernatant was used for enzymatic assays. A working solution was mixed at a 4:1 ratio (R1:R2) and stored at 4 °C. The TG standard was diluted to 6 different concentrations with distilled water to prepare a standard curve, and the control tube contained no TG. All reactions were incubated at 37 °C for 10 min, after which the OD value of each tube was measured.

The lactic acid concentration was calculated based on the protein concentration per mg. A total of 2 × 10^5^ cells in 12-well plates were treated with conditioned culture medium containing 10% dialyzed serum for 24 h. Due to the excretion of lactate from the cells into the medium, aliquots of 100 μL of supernatant were collected by centrifugation at 0 and 24 h. The lactic acid content in the medium was read with an M-100 automatic biosensor/analyzer (Shenzhen Siemen Technology Co., Ltd, China) and standardized. Standard curves of the excretion or uptake of lactate by the cells per hour were plotted. All data were normalized by comparison with the control.

### 2.6. Endogenous or Exogenous Immunoprecipitation Assay

A total of 1 × 10^7^ cells were treated with RIPA lysis buffer containing a protease inhibitor cocktail (4693116001, Roche, 1:200) and a phosphatase inhibitor (P5641, Sigma-Aldrich, 1:100, Darmstadt, Germany) on ice for 30 min, and the supernatant was collected after centrifugation at maximum speed for 30 min at 4 °C. The total protein concentration in a small amount of lysate (input) was determined for Western blot analysis. One to two micrograms of primary antibodies were added to the remaining lysate and incubated for 6 h at 4 °C with slow shaking; these antibodies included anti-lamin A/C rabbit polyclonal antibody (10298-1-AP, Proteintech, Wuhan, China), anti-phospho-lamin A/C (S22) antibody (2026, CST), anti-phospho-lamin A/C (S392) antibody (ab58528, Abcam), anti-AMPK alpha 1 antibody (5831, CST, Danvers, MA, USA), and anti-phospho-AMPK alpha (T172) antibody (2535S, CST). Then, 50 μL of pretreated Protein A/G agarose beads (P2055, Beyotime Biotechnology, Shanghai, China) were added to the cell lysate and incubated overnight with the antibody complex with slow shaking at 4 °C. After the immunoprecipitation reaction, the agarose beads were washed using wash buffer at least 5 times at 4 °C. Finally, the beads were resuspended in 2× SDS loading buffer and boiled at 95 °C for 10 min. Western blotting was performed to analyze protein interactions.

For the exogenous immunoprecipitation assay, plasmids expressing the N-terminal region of AMPKα (1–312, AMPKα-N), the C-terminal region of AMPKα (331–359, AMPKα-C) and full-length AMPKα (AMPKα-F) were constructed with a vector (pCDHO-puro-CMV-3Flag, GENEWIZ, Suzhou, China). HEK293T cells (5 × 10^6^) were transfected with PEI and 10 μg of plasmids. After 48 h of transfection, the nuclei were separated with nucleoplasm separation buffer (A BioVision, #K266-25, Milpitas, CA, USA) and lysed with cold RIPA buffer for 2 h and then the lysate of the nucleus was collected. Thirty microliters of the nuclear lysate supernatant were collected as the input. The remaining nucleus lysate was incubated with anti-FLAG beads (A2220, Sigma–Aldrich, Darmstadt, Germany) at 4 °C with rotation overnight. The beads were washed three times with cold RIPA buffer. The final immunoprecipitants were boiled in 5× SDS loading buffer for immunoblotting.

### 2.7. Tissue and Cell Protein Extraction, BCA Assay and Western Blot Analysis

RIPA lysis buffer and extraction buffer were applied to extract protein from homogenized tissues or cell samples (89900, Thermo Scientific™, Waltham, MA, USA). A BCA kit (Beyotime, P0011, Jiangsu, China) was used to quantify the protein concentrations, and the proteins were then separated on SDS–PAGE gels. Protein fragments were transferred onto a nitrocellulose (NC) membrane and subjected to immunoblot analyses with the indicated primary antibodies at the following dilutions: anti-lamin A/C (10298-1-AP, Proteintech, Wuhan, China) at 1:1000, anti-phospho-lamin A/C (S22) (2026, CST) at 1:1000, anti-phospho-lamin A/C (S392) (ab58528, Abcam) at 1:1000, anti-AMPK alpha 1 (5831, CST) at 1:1000, anti-phospho-AMPK alpha (T172) (2535S, CST) at 1:1000, anti-acetyl-CoA carboxylase (21923-1-AP, Proteintech) at 1:1000, anti-phospho-acetyl-CoA carboxylase (S79) (3661S, CST) at 1:1000, anti-fatty acid synthase (FASN) rabbit polyclonal antibody (10624-2-AP, Proteintech) at 1:1000, anti– mitochondrial pyruvate carrier (MPC1; 14462, CST) at 1:1000, anti-β-hydroxy-β-methylglutaryl coenzyme A reductase (HMGCR) polyclonal antibody (13533-1-AP, Proteintech) at 1:1000, anti-GAPDH (60004-I-Ig, Proteintech) at 1:5000, anti-β-actin (60008-1-Ig, Proteintech) at 1:5000, and anti-FLAG (80010-1-RR, Proteintech) at 1:5000, which was used as a control. The secondary antibodies used were HRP-conjugated goat anti-mouse IgG (H+L) (RS0001, ImmunoWay, Beijing, China) diluted 1:10,000 and anti-mouse IgG (sc-2004, Santa Cruz, Texas, USA) diluted 1:10,000. The membranes were developed using an enhanced chemiluminescence HRP substrate (WBKLS0500, Millipore Corporation Billerica, Billerica, MA, USA) and then exposed to a MiniChemi 610 imager (304002L, SAGECREATION, Beijing, China).

### 2.8. Extraction, Culture, and Induction of AD-MSCs

Fresh mouse adipose tissue was digested with 0.075% collagenase (C0130, Sigma, Darmstadt, Germany) at 37 °C for 30 min, filtered with a 100-μm Falcon cell strainer and centrifuged at 1600 rpm for 8 min. The precipitate was resuspended in ammonium chloride for erythrocyte lysis and incubated at room temperature for 5–10 min. The ammonium chloride erythrocyte lysis buffer (storage solution, 10×) was prepared in a total volume of 1 L: 80.2 g of NH_4_CI (15 M), 8.4 g of NaHCO_3_ (100 mM), and 3.7 g of disodium EDTA (10 mM) in 900 mL distilled water, with the pH adjusted to 7.4. Finally, we added water to bring the volume to 1 L and stored the solution at 4 °C for six months. The lysate was centrifuged at 3000 rpm for 3 min, and the precipitate was washed twice with PBS and 5% serum. The purified AD-MSCs were cultured in 6-well plates in F12 medium (SH30023.01, HyClone, Thermo Fisher Scientific, Waltham, USA) containing 10% serum (16000044, Gibco) and subjected to Oil Red O staining (G1262, Solarbio, Beijing, China) as a WT control and photographed before the induction of differentiation (at day 0). When the cell density reached 80–90%, we replaced the medium with induction differentiation medium A (ready-to-use). Induction differentiation medium A (500 mL) contained the following components: 0.2 mL of Dex storage solution (1 µM), 0.5 mL of IBMX storage solution (0.5 mM), 0.556 mL of a 4.5 mg/mL insulin solution (5 µg/mL), 0.25 mL of Trog storage solution (5 µM), and 500 mL of complete DMEM/F-12. After 48 h of cell culture (day 2), the cell morphology changed, and induction differentiation medium A was slowly exchanged with induction differentiation medium B (ready to use); the cells were cultured for an additional 48 h. Induction differentiation medium B (500 mL) contained the following components: 0.556 mL of a 4.5 mg/mL insulin solution (5 µg/mL), 0.25 mL of Trog storage solution (5 µM), and 500 mL of complete DMEM/F-12. After the cells had been cultured for 96 h (day 4), lipid droplet formation was observed, and the medium was changed to complete F12 medium every two days until the cells became adipocytes. Oil red O staining was performed on the eighth day for identification.

### 2.9. High Glucose-Mediated Increase in Cellular Lipid Synthesis and Oil Red O Staining

Cells were cultured in standard DMEM (glucose concentration of 5.5 mM) supplemented with 10% FBS and 1% streptomycin. After the cells were synchronized by incubation with serum-free medium for 24 h, they were treated with high-glucose DMEM (glucose concentration of 25 mM). Oil Red O staining and TG assays validated intracellular lipogenesis. Specifically, the cells were washed three times with PBS for 5 min and fixed with a 4% paraformaldehyde solution (MFCD00133991, Thermo Scientific™) for 40 min. After the paraformaldehyde was removed, the cells were stained with oil Red O dye for 30 min at room temperature in the dark. Finally, 60% isopropanol was used for rinsing and decolorization. The treated cells were placed under a microscope for observation and photographed. 

### 2.10. Mass Spectrometry

A total of 5 × 10^7^ HEK293T cells were lysed, and nucleoplasmic separation was performed according to the manufacturer’s instructions (BioVision, #K266-25, Milpitas, CA, USA). The endogenous immunoprecipitation method was used to precipitate the nuclear protein complexes interacting with lamin A/C (10298-1-AP, Proteintech). The precipitates were separated on a 4–12% NuPAGE gel (Thermo Scientific, NP0335BOX). Silver staining was performed using a Pierce Silver Stain Kit (Thermo Scientific, 24612). The input was used as a positive control, while IgG was used as a negative control. Protein bands from the Gels were subjected to LC/MS-MS (Agilent 6340, Palo Alto, CA, USA) analysis.

### 2.11. CCK-8 Assay

The proliferation of 3000 cells seeded in each well of a 96-well plate was assessed after different treatments. Ninety-five microliters of medium and 5 μL of CCK-8 reagent (E1CK-000208, EnoGene, Nanjing, China) were mixed and added to the 96-well plate. After an incubation for 1–4 h, the absorbance at 450 nm was determined with a spectrophotometer (Berthold Tristar, 2LB942, Germany).

### 2.12. Statistical Analysis

The Western blot, qPCR, CO-IP and CCK-8 experiments were repeated at least three times. Prism 8.0 software (GraphPad Software, San Diego, CA, USA) was applied for statistical analyses and plotting graphs. Standard deviations (SD) were calculated. Differences between groups were analyzed using a two-tailed Student’s *t*-test, and Pearson’s correlation test was performed to analyze correlations between the levels of different indexes. *p* < 0.05 indicated a statistically significant difference.

## 3. Results

### 3.1. pLamin A/C Levels Increased around Lipid Droplets in Human Liver Tissue

By analyzing lamin A/C or pLamin A/C (S392) levels in human normal fat and liver samples using IHC, we found that pLamin A/C was distributed in the area surrounding lipid droplets in liver tissues, suggesting its function in lipid droplet formation (Figure 1A). Lipid vesicles in the liver tissue are highlighted with a red arrow (Figure 1A, right panel). We established animal models for further validation. Obese Sprague–Dawley (SD) rats (OBE rats) were obtained by continuous high-fat diet feeding for eight weeks. The body weights of some rats fed high-fat food were not significantly higher than of those in the control group (CON); these rats were therefore included in the obesity-resistant group (RES). Rats in the control group were fed the basal diet. We performed a semiquantitative analysis of changes in FASN, ACC1, lamin A/C and pLamin A/C levels in adipose and liver tissues from the rats using Western blotting (*n* = 5) (Figure 1B and Appendix A). Notably, FASN showed a significant downregulation in fat tissues from OBE rats compared with CON rats, but the differences in the levels of other proteins were not significant.

### 3.2. LMNA KD Blocked AD-MSC Maturation, and LMNA Deletion Decreased Lipid Synthesis in 7701 Cells

Furthermore, we isolated AD-MSCs from mice and cultured them. The primary AD-MSCs differentiated into mature adipocytes, which showed deep oil red O staining after eight days following stimulation with a conditioned culture medium that contained insulin and dexamethasone (Figure 2A, the three images on the left). In addition, we detected elevated lamin A/C levels in mature adipocytes (Figure 2B, left panel); in contrast, the maturation and proliferation of AD-MSCs were prevented upon *LMNA* KD (Figure 2A (right panel), Figure 2B (middle panel) and Figure 2C (left panel)). Notably, the deletion of *LMNA* in AD-MSCs cannot survive (data not shown). The data collected from adipose tissue suggest that lamin A/C controls the proliferation and maturation of adipocytes. Fortunately, we obtained 7701 cells in which *LMNA* had been deleted (Figure 2B, right panel); these cells showed significantly decreased TG synthesis but increased lactic acid production (Figure 2D) and reduced proliferation (Figure 2C, right panel). After cell growth was synchronized by overnight serum starvation, we observed elevated pACC1-S79 levels, decreased FASN levels and no change in β-hydroxy-β-methylglutaryl coenzyme A reductase (HMG-CoA reductase, HMGCR) levels in *LMNA*-KO 7701 cells compared to WT 7701 cells (Figure 2E). In classical metabolic pathways, FASN, ACC1 and HMGCR are rate-limiting enzymes for de novo lipid synthesis and cholesterol synthesis under normal conditions [31], and pACC1-S79 is inactivated ACC1 [32], thus explaining the decrease in lipid synthesis in *LMNA*-KO 7701 cells. The levels of the MPC1 and MPC2 proteins, but not mRNAs, were decreased in *LMNA*-KO 7701 cells, similar to ACC1 levels (Appendix A). Consistent with this result, the MPC 1/2 heterodimer in the inner mitochondrial membrane contributes to controlling the acetyl-CoA concentration in the cytosol by transporting pyruvate (a product of anaerobic glycolysis) into mitochondria. Malfunction of MPC1 and MPC2 leading to pyruvate accumulation may induce an increase in lactate levels [33].

### 3.3. Lamin C and Phosphorylation-Activated AMPKα Directly Interacted in 7701 Cells

AMPKα directly phosphorylates ACC1 to inactivate it [34]. As an energy sensor, AMPKα promotes glycolysis via the activation of phosphofructokinase 2 (PFK2) and inhibits lipogenesis via the phosphorylation of its substrate ACC1 [35,36]. In this context, after cell growth was synchronized by overnight serum starvation, we observed elevated pAMPKα-T172 levels (active form) and decreased pAMPKα-S485 levels (inactive form) in *LMNA*-KO 7701 cells compared to WT 7701 cells (Figure 3A,B). Furthermore, we performed immunoprecipitation experiments with endogenous proteins to determine whether lamin A/C interacted with pAMPKα and whether pLamin A/C interacted with FASN, pACC1 or pAMPKα (Figure 3C and Appendix A). The results supported the direct interaction between lamin A/C and pAMPKα-T172 (Figure 3C, highlighted by the red arrow). Through exogenous reverse immunoprecipitation experiments of proteins with a Flag tag, we validated the lamin C-pAMPKα-T172 interaction in 7701 cells (Figure 3D). Furthermore, we found that this direct interaction occurred in the N-terminal kinase domain of AMPKα in nuclear extracts (Figure 3E), however, both N and C termini of AMPKα could interact with lamin C in whole cell lysates (Appendix A). These results strongly suggest that AMPKα in the nucleus might be a key target by which lamin A/C mediates metabolism in liver tissue.

### 3.4. AMPKα Activation Is Closely Associated with Abnormal Lamin A/C Function

We constructed different plasmids containing *LMNA* mutations associated with FPLD [37,38,39,40], transfected them into 7701 cells and assessed changes in metabolic functions and pAMPKα-T172 levels in 7701 cells to understand how *LMNA* deletion activates AMPKα. The schematic diagram in Figure 4A indicates the location of the mutation in the gene and protein sequences. Using Western blotting, we found that the mutant plasmids differentially affected the expression of mature lamin A/C (Figure 4A). Mutation 1 (M1, D230N) in the rod domain and mutation 2 (M2, G465D) in the tail domain (LTD) did not rescue mature lamin A/C expression; however, it did so for both mutation 3 (M3, R482W) and mutation 4 (M4, T528R) in the LTD rescued expression. Coincidentally, the 528 site is threonine phosphorylation site of lamin A/C and the T528R mutation prevents its phosphorylation. Additionally, transfection of the *LMNC* cDNA (572 aa, sc321549) successfully rescued mature lamin C expression, but transfection of the full-length *LMNA* cDNA (664 aa) with GFP tag at the C-terminus rescued prelamin A-GFP fusion protein expression. The increase in pAMPKα-T172 levels in the groups expressing *LMNC* cDNA, M3 or M4 was abrogated (Figure 4B). Prelamin A-GFP or *LMNA* deletion increased the pAMPKα-T172 levels. Furthermore, we examined the functional effects of these changes by measuring TG and lactic acid synthesis. The restored expression of lamin C successfully abrogated the increase in lactate synthesis and decrease in TG synthesis caused by *LMNA* deletion, but M3 and M4 in the LTD only promoted lactate synthesis. Notably, phosphorylation inactivation due to M4 increased TG synthesis (Figure 4B), suggesting that modification of the lamin A tail is closely related to abnormal lipid metabolism and that phosphorylation at T528 plays a role in inhibiting lipid synthesis. Farnesylation of prelamin A contributes to the transport of prelamin A to the nuclear membrane [41]. Lonafarnib was used to inhibit the farnesylation of prelamin A and validate the transient activation of AMPKα associated with abnormal prelamin A maturation. As expected, the loss of mature lamin A/C activated AMPKα through phosphorylation, and activated AMPKα (phosphorylated AMPKα) subsequently targeted ACC1, leading to intracellular changes in metabolic homeostasis in the three liver cell lines (Figure 4C).

### 3.5. LMNA Deletion Activates AMPKα, Possibly via a Change in the ADP/ATP Ratio

We performed endogenous IP of lamin A/C in the nucleus and identified enzymes interacting with lamin A/C that are closely related to ATP synthesis, such as transitional endoplasmic reticulum ATPase and ADP/ATP translocase 2, using a mass spectrometry analysis to understand the mechanism by which AMPKα is activated (Figure 5A,B, Table 1). Lamin A/C may be involved in regulating ATP metabolism in the nucleus; however, the ADP/ATP ratio is a direct factor contributing to AMPKα activation. Does pAMPKα-T172 phosphorylate lamin A/C? After three cell lines were treated with a selective AMPK activator (A-769662) or an AMPK inhibitor (compound C), we found that both the inhibition and activation of AMPKα resulted in decreased levels of lamin A/C and pLamin A/C-S22 (Figure 5C). Recent reports have revealed that A-769662 inhibits adipocyte differentiation, proteasomal activity, cell growth and DNA replication via an AMPK-independent mechanism [42]. As an important molecule for differentiation, lamin A/C is presumably a direct target of A-769662.

### 3.6. pLamin A/C Was Upregulated and pAMPKα-T172 Levels Were Decreased during High Glucose-Induced Lipid Synthesis

Based on the results of the mass spectrometry analysis, lamin A/C binds not only to fatty acid-binding protein 5 (FABP5) but also to pyruvate kinase (PKM), suggesting that lamin A/C is an important regulatory factor maintaining the homeostasis of glycolipid metabolism. We assessed this effect under physiological conditions in three cell models treated with high glucose (25 mM) for 12 h and found that lamin A/C phosphorylation and TG synthesis were increased (Figure 6A,B). FASN expression was increased in two hepatoma cell lines (Figure 6A). Additionally, pLamin A/C expression in HepG2 cells exhibited a significant time-dependent increase after stimulation with 25 mM glucose (Figure 6C). Thus, pLamin A/C is an indicator of increased lipid metabolism and synthesis. In contrast, the level of pAMPKα-T172 was reduced in 7701 and HepG2 cells, indicating that decreased AMPKα activity promotes lipid synthesis under high-glucose conditions (Figure 6D). According to the correlation analysis, pLamin A levels showed a significant positive correlation with FASN expression (R^2^ = 0.26, *p* = 0.04) and a trend toward a negative correlation with pAMPKα-T172 levels (R^2^ = 0.3, *p* = 0.05, Figure 6E).

### 3.7. Combined Targeting of AMPKα and Lamin A/C Effectively Inhibited the Growth of HCC Cells

Liver cancer is strongly associated with abnormal lipid metabolism [43]. According to an analysis of TCGA database, *LMNA* expression gradually increased with the liver cancer grade, consistent with the findings from HCC cell lines with different degrees of malignancy (Figure 7A,B). Patients with HCC presenting higher LMNA levels experienced shorter survival than those with low/medium *LMNA* levels (Figure 7A, right panel). Neither HepG2 nor MHCC97-H cells survived for more than three days upon *LMNA*-KD. These findings indicate that *LMNA* acts as an oncogene in HCC. We also found that pAMPKα-T172 levels increased in a concentration-dependent manner when HepG2 or MHCC97-H cells were treated with lonafarnib (Figure 4C). In addition, lonafarnib reduced the expression of HMGCR, the rate-limiting enzyme in cholesterol synthesis, in a concentration-dependent manner (Figure 7C). Cholesterol is an important energy substrate for the metabolism of liver cancer cells [44]. However, because pAMPKα-T172 promotes glycolysis to maintain energy in cancer cells, the synergistic use of lonafarnib and glucose-lowering therapies should more efficiently inhibit the growth of HCC cells. As expected, 4 μM lonafarnib combined with 250 μM metformin significantly inhibited the growth of HepG2 and MHCC97-H cells compared with their growth observed upon treatment with lonafarnib alone (Figure 7D).

## 4. Discussion

Acetyl coenzyme A (acetyl-CoA) regulates metabolic flux. Excess acetyl-CoA from a high-glucose diet enters de novo lipid synthesis and cholesterol synthesis, and these reactions occur mainly in liver tissues or adipose tissues [45]. In our study, IHC staining of human liver and adipose tissues revealed that lamin A/C localized to the nuclear membrane and that pLamin A/C was distributed mainly in the nucleoplasm and around lipid droplets. However, we did not observe abnormal expression of lamin A/C or pLamin A/C in the adipose tissue of obese rats. Some earlier publications stated that lamin A/C plays a role in promoting differentiation, especially in mesenchymal stem cells and even in cancer cells [14,46]. Consistent with this function, our experiments confirmed that *LMNA* KD in AD-MSCs of mice hindered adipocyte development and proliferation and subsequently reduced lipid synthesis. Notably, we observed an increasing trend in pLamin A/C-S392 expression in the liver tissue of obese rats, suggesting that pLamin A/C regulates fat metabolism in liver tissue. Using hepatocyte lines (7701 cell line), we identified new functions of lamin A/C linked to energy metabolism and a new role for pLamin A/C in de novo lipid synthesis that has not yet been reported in the literature.

The NetPhos 3.1 server predicted that lamin A has 125 phosphorylation sites and that pLamin A/C-S392 has kinase activity. The extra function of lamin A/C on different phosphorylation sites is rarely reported, except for “mitotic sites” at S22 or S392 [10]. We investigated the mechanisms by which pLamin A/C (S22 or S392) coordinates metabolic pathways in the liver, the central metabolic organ to explain the visceral fat deposition and metabolic complications in *LMNA*-linked FPLD, including hyperglycemia, hypertriglyceridemia and fatty liver [47]. We concluded that pLamin A/C at S22 or S392 participates in TG metabolism. In 7701, HepG2 and MHCC97-H cells stimulated with high glucose, pLamin A/C-S22/392 levels increased along with increased TG synthesis. This change in pLamin A/C-S392 levels positively correlated with the change in FASN levels. However, TG synthesis was significantly decreased in the *LMNA*-deleted 7701 cell line stimulated with high glucose. The mechanism of pLamin A/C-S392 involvement in lipid synthesis in the nucleus deserves further study in the future.

Unexpectedly, we observed increased pAMPKα-T172 levels in stable *LMNA*-KO 7701 cells, accompanied by a decrease in TG synthesis and an increase in lactate levels. AMPK, which belongs to the Ser/Thr protein kinase family, contains three catalytic subunits: α, β and γ. AMPKα inhibits lipid synthesis via direct serine phosphorylation of ACC1-S79, which promotes glycolysis and fatty acid oxidation, but high glucose levels may inhibit AMPKα activation [48]. AMPKα activation was reported to alter *LMNA* splicing and thus alleviate progerin production [49]. Our experiment showed that lamin A/C precipitated the pAMPKα-T172 protein in 7701 cells, but pLamin A/C-S392 did not. Based on the Human Protein Atlas, AMPKα expressed in single human cells mainly localizes to nuclear speckles [50]. In addition, it localizes to the cytosol. The lamin A/C protein is the principal protein in nuclear speckles [51]. In the absence of the interaction between pLamin A/C-S392 and pAMPKα-T172, AMPK activation should correlate with lamin A/C processing. We inhibited prelamin A maturation using farnesylation inhibitors to further validate this hypothesis and detected increased pAMPKα-T172 levels in 7701, HepG2 and MHCC97-H cells. This phenomenon is prominent in cancer cells. Lamin A/C is necessary for the survival of HepG2 and MHCC97-H cells, thus primarily establishing an association between lamin A/C and AMPK pathways and hepatocellular carcinoma. Notably, an AMPK inhibitor and two AMPKα activators, A769662 and AICAR (Appendix A), decreased lamin A/C and pLamin A/C levels, indicating that lamin A/C is not a substrate of pAMPKα. Additionally, we found that deletion of *LMNA* in 7701 cells downregulated the expression of MPC1 and 2, possibly resulting in increased glycolysis and lactate accumulation (Appendix A), which requires further investigation in the future.

By performing mass spectrometry of endogenous proteins in HEK293T cells, we identified that some proteins recruited by lamin A/C are functionally enriched in keratinocyte migration, protein refolding, intermediate filament cytoskeleton organization, canonical glycolysis, NADH metabolic process, and positive regulation of ATP biosynthetic process, suggesting that lamin A/C functions in regulating ATP metabolism. AMPKα activation coordinates glycolysis and fat synthesis to maintain energy homeostasis. However, pLamin A/C is independent of the AMPKα pathway and is mainly involved in TG anabolism.

Laminopathies are a spectrum of diseases caused by *LMNA* mutations. The ClinVar database contains a summary of many FPLD-associated *LMNA* mutations [38,39,40]; among them, D230N in the α-helical rod domain and R482W, G465D, and T528R in the LTD were selected to analyze the effect on pLamin A/C-S392 in 7701 cells. The function of lamin A/C phosphorylation at site 528 has not been reported. When these mutant plasmids and control plasmids were transfected into the *LMNA*-KO 7701 cell line, the plasmid containing the mature *LMNC* cDNA successfully abrogated the increase in pAMPKα-T172 in the KO group; however, the prelamin A-GFP did not. Further confirmation showed that pAMPKα activation was associated with the blockade of lamin A/C maturation. Mutations at codons 230 and 465 of *LMNA* resulted in the loss of lamin A/C protein expression and decreased TG synthesis, similar to the effects of *LMNA* KO; however, significant activation of AMPKα was not observed. This finding reveals the important role of pLamin A/C in lipid synthesis. Following rescued expression of lamin A upon its mutation at codons 482 and 528, the increase in pAMPKα-T172 was alleviated. The T528R mutation caused an increase in the synthesis of both lactate and lipids in 7701 cells; however, R482W only increased lactate synthesis. Thus, lamin A/C proteins phosphorylated at different sites exert different effects on lipid metabolism.

Numerous studies have reported a link between aberrant lipid metabolism and liver cancer. Lamin A/C overexpression is associated with the malignancy of liver cancer and patient survival. *LMNA* functions as an oncogene in HCC. Therapeutic strategies targeting *LMNA* mutations or abnormal lamin A/C expression have recently been accepted. Lonafarnib has been approved by the FDA for the treatment of hepatitis D virus (HDV) infection, progeria and progeroid laminopathies [52,53]. Some animal experiments have also validated the efficacy of lonafarnib alone for the treatment of HCC [53]. Here, we found that the combination of glucose restriction and targeting lamin A/C increased the efficiency of HCC treatments. This finding requires further validation in clinical trials.

## 5. Conclusions

Lamin A/C distributed in the nuclear membrane coordinates energy metabolism in the nucleus. Under normal conditions, lamin A/C promotes lipid synthesis through the pLamin A/C pathway in liver cells. *LMNA* loss in the nuclear membrane promotes glycolysis and inhibits fat synthesis via AMPKα phosphorylation at site 172. The mutation-induced disruption in the maturation of lamin A/C in the nucleus potentially interferes with glycolipid metabolic homeostasis in liver cells.

## Figures and Tables

**Figure 1 cells-11-03988-f001:**
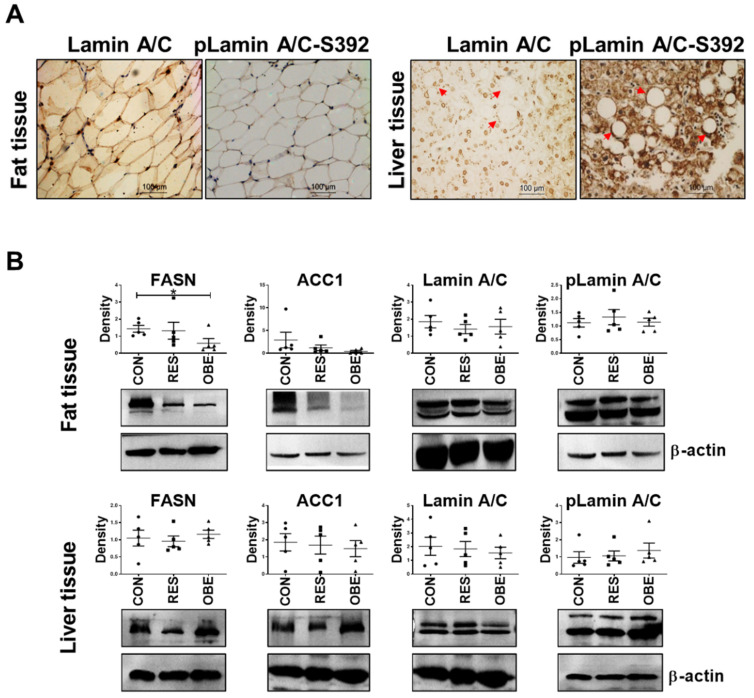
The pLamin A/C aggregates around lipid droplets. Obesity was induced in SD rats by continuously feeding animals a high-fat diet for eight weeks. (**A**) Representative images of IHC staining for lamin A/C and pLamin A-S392 in human adipose and liver tissues (20×). The red arrow indicates pLamin A/C concentrated around lipid droplets in liver tissues (*n* = 3). (**B**) Representative Western blot images showing FASN, ACC1, lamin A/C or pLamin A/C levels in adipose and liver tissues from OBE rats and the corresponding β-actin showing the quantitative analysis (*n* = 5). Data are presented as the means ± SD. * *p* < 0.05.

**Figure 2 cells-11-03988-f002:**
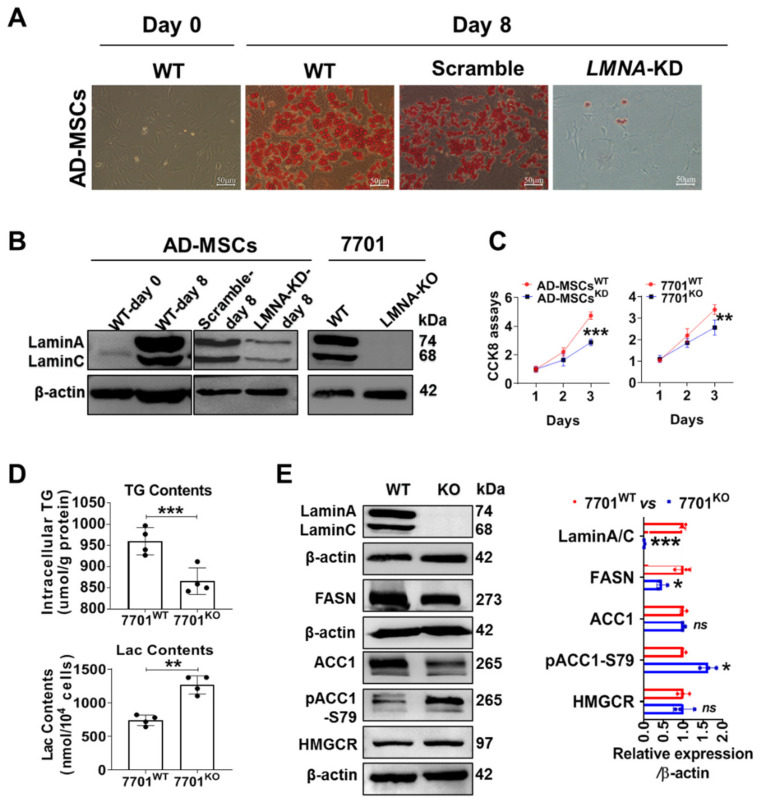
Lamin A/C has different functions in metabolism in hepatocytes and adipocytes. (**A**) Oil red O staining was performed to show TG synthesis in primary cultured AD-MSCs or AD-MSCs after differentiation was stimulated for eight days or in stable *LMNA*-KD cells generated via CRISPR–Cas9 (*n* = 3). (**B**) Western blot analysis of the expression of lamin A/C under different conditions. *LMNA* was stably deleted from 7701 cells, as validated by Western blotting. (**C**) CCK-8 assays showed the inhibition of the growth of *LMNA*-KD AD-MSCs or *LMNA*-KO 7701 cells compared with scrambled shRNA-expression or WT cells (*n* = 8). (**D**) TG and lactate synthesis were quantified in 7701 cells (*n* = 4). (**E**) The expression levels of key enzymes and proteins involved in de novo fatty acid synthesis in *LMNA*-KO 7701 cells are shown along with the corresponding plot of the quantitative analysis (*n* = 3). Data are shown as the means ± SD. *ns*, *p* > 0.05, * *p* < 0.05, ** *p* < 0.01, and *** *p* < 0.001.

**Figure 3 cells-11-03988-f003:**
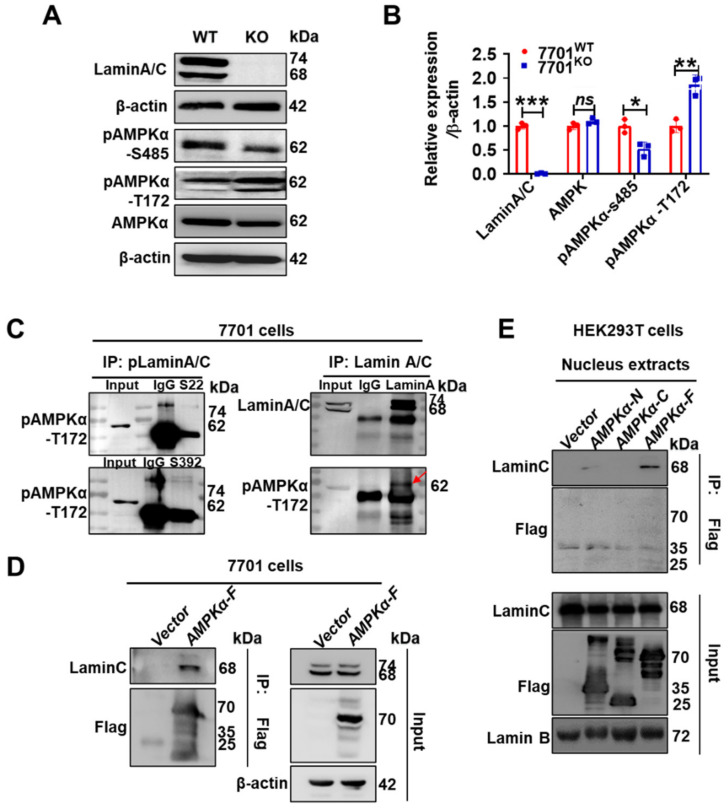
*LMNA* deletion inhibits de novo fat synthesis through AMPKα phosphorylation. (**A**) The expression levels of AMPKα and its phosphorylated form in *LMNA*-KO 7701 cells are shown with the corresponding plot of the quantitative analysis (*n* = 3) (**B**). (**C**) CO-/C IP assays were performed to assess the endogenous interaction between pLamin Aand pAMPKα-T172 or between lamin A/C and pAMPKα-T172 in 7701 cells. IgG was used as a negative control, and the input was used to examine the levels of pAMPKα-T172 and lamin A/C. (**D**) An exogenous reverse pull-down experiment was performed in 7701 cell lines. 7701 cells were transiently transfected with the *AMPK*α-FLAG plasmid or control vector for 48 h. An anti-FLAG antibody was used to immunoprecipitate the cell lysates. (**E**) HEK293T cells were transiently cotransfected with the plasmid containing the *LMNA* cDNA and the *AMPKα*-N, -C and -F plasmid or control vector. An anti-FLAG antibody was used to immunoprecipitate the nuclear lysates. Data are presented as the means ± SD. *ns*, *p* > 0.05, * *p* < 0.05, ** *p* < 0.01, and *** *p* < 0.001.

**Figure 4 cells-11-03988-f004:**
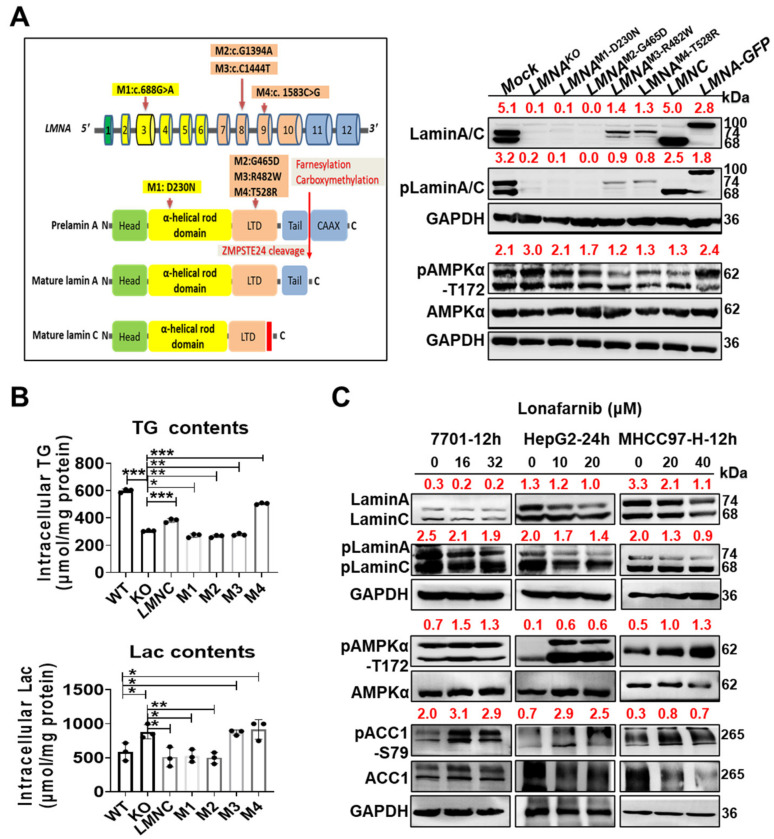
Abnormalities in Lamin A/C function trigger AMPKα activation. (**A**) Mutations in the *LMNA* structure and sequence are shown: D203N is located in the rod domain and R482W, G465D, and T528R are located in the LTD. Western blotting was performed to detect changes in lamin A/C, pLamin A/C, AMPKα, and pAMPKα-T172 levels upon the rescue of lamin A/C expression via the transfection of the *LMNC* cDNA, *LMNA-GFP* cDNA and *LMNA* mutants, and the corresponding semiquantitative data obtained after GAPDH normalization are shown. (**B**) TG and lactate synthesis were quantified in the different transfected groups (*n* = 3). (**C**) Western blotting was performed to assess concentration-dependent changes in lamin A/C, pLamin A/C, AMPKα, pAMPKα, pACC1 and ACC1 levels in 7701, HepG2 and MHCC97-H cells treated with lonafarnib. The corresponding quantitative data were obtained by standardization with GAPDH. Data are presented as the means of at least three independent experiments ± SD. * *p* < 0.05, ** *p* < 0.01, and *** *p* < 0.001.

**Figure 5 cells-11-03988-f005:**
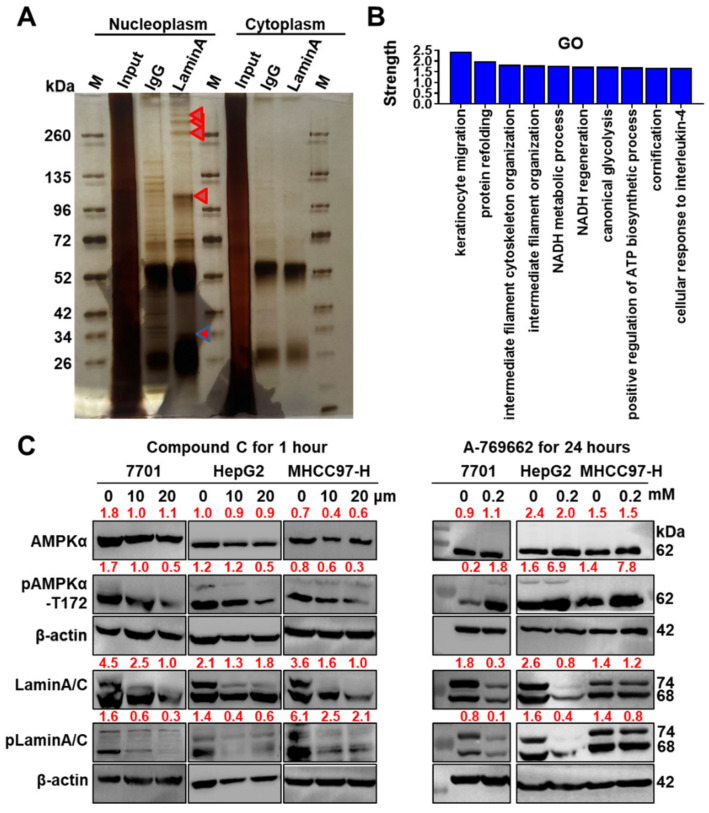
Lamin A/C is not a substrate of pAMPKα, and ATPase is one partner of lamin A/C. (**A**) Mass spectrometry analysis of intranuclear immunoprecipitants interacting with lamin A/C. Silver-stained gel showing isolated fragments of immunoprecipitants; red triangles show target bands used for the mass spectrometry analysis. (**B**) Online STRING analysis of the GO terms of the top 30 proteins identified using mass spectrometry. The top 10 GO terms are listed. (**C**) Left panel: AMPKα activation was inhibited by compound C in a concentration-dependent manner. Right panel: When AMPK was activated with A-769662, lamin A/C and pLamin A/C levels were still decreased in 7701, HepG2 and MHCC97-H cells. Semiquantitative values from the blots are displayed.

**Figure 6 cells-11-03988-f006:**
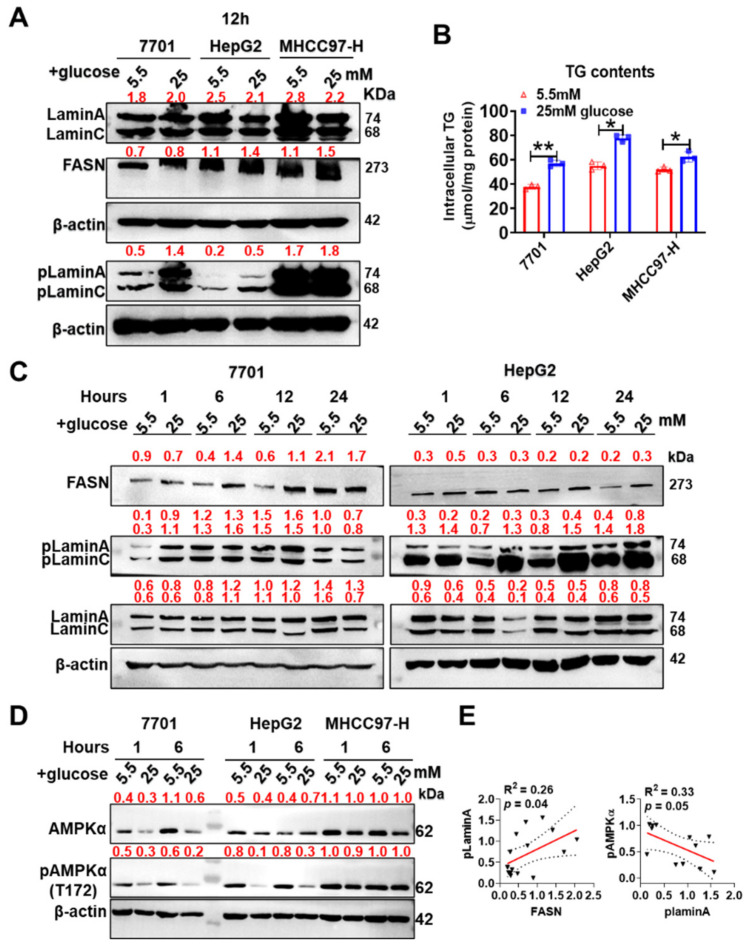
pLamin A/C is an indicator of increased lipid synthesis. (**A**) Treatment with 25 mM glucose for 12 h promoted lamin A/C phosphorylation and FASN expression in 7701, HepG2 and MHCC97-H cells. Semiquantitative analyses of blots are displayed. (**B**) TG synthesis was quantified (*n* = 3). (**C**) Lamin A/C phosphorylation was increased by 25 mM glucose in a time-dependent manner in 7701 and HepG2 cells. Gray values from blots are displayed. (**D**) The inhibition of AMPKα activation was shown in 7701, HepG2 or MHCC97-H cells treated with 25 mM glucose. Semiquantitative analyses of blots are displayed. (**E**) Person’s correlation analysis between pLamin A and FASN or pAMPKα-T172 is shown. Data are shown as the means of at least three independent experiments ± SD. * *p* < 0.05 and ** *p* < 0.01.

**Figure 7 cells-11-03988-f007:**
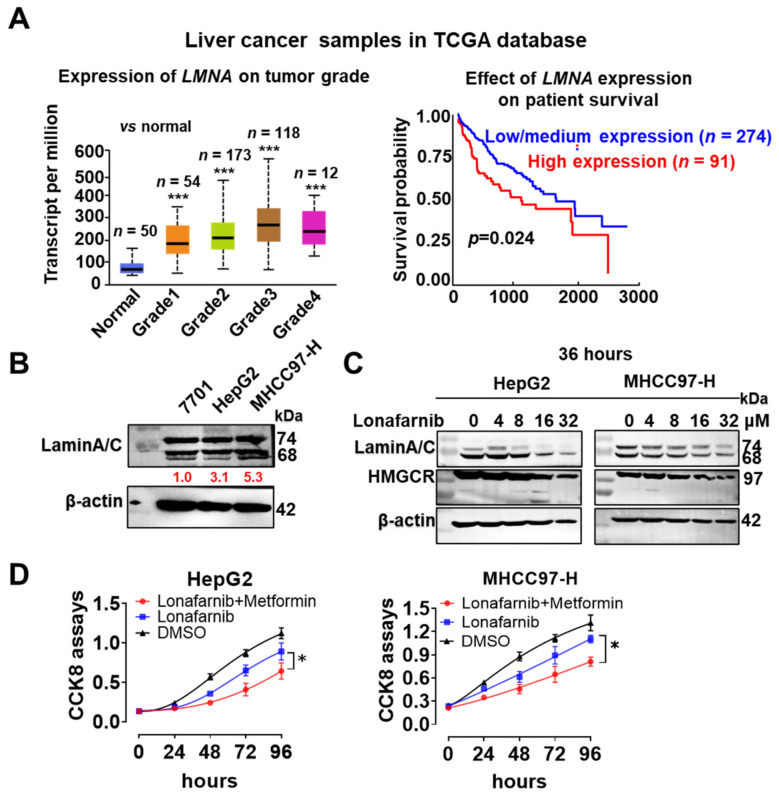
Lonafarnib combined with metformin significantly inhibited the growth of HepG2 and MHCC97-H cells. (**A**) Left panel: *LMNA* expression gradually increased with liver cancer grade based on data from TCGA database (*n* = 407). Right panel: The survival of patients with high *LMNA* levels was significantly shorter than that of patients with low/medium *LMNA* levels. (**B**) Lamin A/C levels in HCC cell lines with different degrees of malignancy and the corresponding gray value (*n* = 3). (**C**) Western blotting showed that lonafarnib reduced the expression of lamin A/C and HMGCR in a concentration-dependent manner. (**D**) CCK-8 assays were performed to assess the growth of cancer cells treated with 4 μM lonafarnib and 250 μM metformin (*n* = 6). The dose of lonafarnib alone was 16 μM. Three independent experiments were performed. Data are presented as the means ± SD. * *p* < 0.05 and *** *p* < 0.001.

**Table 1 cells-11-03988-t001:** Mass spectrometry detected partners of nuclear lamin A/C.

Peptides	PSMs ^1^	Group Description
32	279	Prelamin-A/C (LMNA)
10	61	Lamin-B1 (LMNB1)
19	36	Keratin, type II cytoskeletal 1 (KRT1)
12	20	Keratin, type II cytoskeletal 2 epidermal (KRT2)
7	20	Tubulin alpha-1B chain (TUBA1B)
9	19	Keratin, type I cytoskeletal 9 (KRT9)
6	16	Tubulin alpha-1C chain (TUBA1C)
4	16	Lamin-B2 (LMNB2)
8	13	Heat shock protein HSP 90-beta (HSP90AB1)
6	13	Tubulin beta chain (TUBB)
7	12	Heat shock 70 kDa protein 1B (HSPA1B)
7	12	Actin, cytoplasmic 1 (ACTB)
8	11	Keratin, type I cytoskeletal 10 (KRT10)
7	11	Keratin, type II cytoskeletal 6B (KRT6B)
5	10	Elongation factor 1-alpha 1 (EEF1A1)
1	10	Serum albumin (ALB)
8	9	Keratin, type II cytoskeletal 5 (KRT5)
7	8	Keratin, type II cytoskeletal 6C (KRT6C)
4	8	Heterogeneous nuclear ribonucleoprotein H (HNRNPH1)
6	7	Nucleolar RNA helicase 2 (DDX21)
7	7	Keratin, type I cytoskeletal 16 (KRT16)
4	7	Alpha-enolase (ENO1)
4	7	Heat shock cognate 71 kDa protein (HSPA8)
5	6	Pyruvate kinase (PKM)
4	6	Elongation factor 2 (EEF2)
5	5	T-complex protein 1 subunit alpha (TCP1)
3	3	Transitional endoplasmic reticulum ATPase (VCP)
2	2	ADP/ATP translocase 2 (SLC25A5)
1	1	Fatty acid-binding protein 5 (FABP5)
1	1	Proliferation-associated protein 2G4 (PA2G4)

^1^ PSMs is the abbreviation of the peptide–spectrum matches.

## Data Availability

All data generated or analyzed during this study and its supplementary information files are included in this published article. Cell lines and most reagents generated for this study are available upon request.

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
