# Peer review of "Loss of Mature Lamin A/C Triggers a Shift in Intracellular Metabolic Homeostasis via AMPKα Activation"

_cells, 2022, doi:10.3390/cells11243988_

Round 1

Reviewer 1 Report

The manuscript addresses a new and interesting issue related to lamin A/C -AMPK relationship in metabolic disorders. The study is well designed, although several concerns raise that need to be addressed.

1) It is really surprising that the authors find lamin A/C around lipid vesicles in liver tissue. To make this finding convincing, a better staining is needed, showing lamin A/C (not only pLMNA) around lipid vesicles and counterstaining nuclei in each picture (Figure 1). 

2) The authors do not clearly define molecules they used in the study as well as antibodies used to detect proteins and even plasmids. Please, call lamin A as lamin A and not LMNA (which is referred to the gene), lamin C alone or lamin A/C.

3) In my opinion it is hard to say that differentiation is impaired if also proliferation and even survival are impaired. Please, consider revising this aspect of the study. (Lines 272-274 and figure 2).

4) In the co-IP in figure 3E (referred to as F in the manuscript?) it is really hard to understand which plasmid has been used to transfect each sample. On the other hand, the co-IP in Figure 3D between LMNA and pAMPKa is convincing, but again, what does LMNA mean? Lamin A, Lamin A/C, prelamin A included?

5) In figure 4, it is surprising that the authors do see endogenous lamin A and C in WT cells (or is it a plasmid expressing both lamin A and C?) and only lamin A in M3 and M4 and neither lamin A nor lamin C in M1 and M2.

6) Data on LMNA upregulation (lines 254-257) are not convincing and I suggest to remove those results.

As a whole, I think that an improvemnt of the way data are presented can provide a good manuscript, yet a great improvement is needed. 

Reviewer 2 Report

The authors reported that LMNA regulates energy metabolism through pLMNA or AMPKα phosphorylation and could be a cancer treatment target. Those findings are novel and of interest to this journal. However, before publishing, the authors should address several problems.

Major

1. In Figure1A, how was the character of the person those tissue samples were derived from? Obese, Normal, or having fatty liver? Please provide the information if the authors know about the person’s state.

2. In Figure 1B, FASN does not seem to increase in OBE in Liver tissue. Instead, it showed a decrease in Fat tissue. The authors state in the manuscript that it is upregulated. This is not correct and confusing.

3. The authors show pLMNA-S392 surrounding lipid droplets in the human liver. How was that in OBE rat? Did the authors also observe pLMNA-S392 surrounding lipid droplets? What is the rationale for pLMNA to surround lipid droplets?

4. The IHC of pLMNA-S392 seems to have high background intensity and looks non-specific. Or another area seems to have a higher signal of pLMNA. Is this signal specific? If so, which part of the liver tissue has the strongest intensity? Again, please provide the information with positive control.

5. How did the deletion of LMNA from AD-MSCs affect survival? Please state that clearer in the manuscript.

6. In IP-western of Figure 3, the pAMPKα-T172 band seems non-specific. How did the authors believe this signal is specific? And in Figure 3E, the signal intensity of β-actin is different also in Input. This signal should be at least the same in the four samples.

7. The authors stated in line 348 that phosphorylation at T528 plays a role in inhibiting lipid synthesis. Where did the authors discuss the phosphorylation position on LMNA? Do they mean mutation at T528?

8. In Figure 5C, AMPK inhibitor Compound C treatment seems to increase Lamin A/C in the 7701 cell line. How did this occur? Is this compound just affect pAMPKα-T172? How about the influence on pAMPKα-S485?

9. About the TCGA database analysis, the authors showed the LMNA expression increased with liver cancer grade. Where does the LMNA expression increase? In the liver or the blood, or any other organ?

10. The authors stated that pAMPK-T172 levels increase with lonafarnib treatment, but Figure 7C doesn’t contain the result. Please provide the result.

11. It’s reasonable for the cancer cell to increase LMNA for its proliferation. On the other hand, glycolysis should be promoted for its energy demand. But as the authors found, glycolysis is activated under AMPKα activation in LMNA KO cells. It’s conflicting. How do the authors understand the higher LMNA expression in cancer?

12. What is the mechanism of AMPKα-T172 phosphorylation under LMNA KO or inhibition of prelaminA farnesylation?

13. This reviewer is afraid of the difficulty of understanding this manuscript to understand for the readers. Therefore, it’s better to make a schematic diagram of the relationship of the molecules stated in this paper.

Minor

1. The authors should spell out the abbreviations at the first appearance also in the manuscript.

2. In Figure 1 legend, the length of feeding a high-fat diet differs from that in the manuscript. Which is correct?

3. In the left lower panel of Figure 3D, the numbering of band size seems different from the left upper panel. The pAMPKα-T172 band should appear in 62 kDa. Probably the numbering misses the proper weight.

4. Figure 5A in line 377 should be Figure 5C.

5. In the right panel of Figure 6E, “plaminA” should be “pLaminA.”

6. In line 420, the word order should be “the growth of HCC cells”

7. The authors use pAMPKα-T172 in some parts and pAMPK-T172 in other parts. Those should be consolidated. It’s same for LMNA and LaminA, and pLMNA and pLaminA.

Reviewer 3 Report

In the submitted manuscript entitled “Loss of mature LMNA triggers a shift in intracellular metabolic homeostasis via AMPKα activating,” the authors address the molecular link between LMNA and pLMNA in regulation of lipid metabolism and show that active form of AMPK (AMPKα-T172) directly interacts with LMNA. In addition, they have reported an efficient way to inhibit cell growth in hepatocellular carcinoma cells by targeting LMNA in combination with glucose restriction in HepG2 and MHCC97-H cell lines. Overall, the submitted manuscript makes some important contributions to the field of laminopathies, liver biology, and lipid metabolism. However, there are a few concerns that should be addressed:

Major comments: 

11. In Figure 1, LMNA and pLMNA localization are determined in human adipose and liver tissue, while their expression are determined via western blot in rat samples. It is not clear to this reviewer that the primary claim of this figure (that pLMNA is elevated in lipogenesis) is justified; it does not appear that such a conclusion can be drawn from the data presented, particularly in part B. A better understanding of the possible role(s) of LMNA and pLMNA with regard to lipid droplets might be obtained via staining of rat tissues, as was done for human tissues in part A. Inclusion of control tissues would be of particular value here, as it may be that pLMNA distribution changes with increased lipogenesis even if its overall level does not change. Relatedly, it was previously reported that hepatocyte-specific deletion of lamin A/C induced spontaneous liver injury specifically in male mice (CMGH, 2017). Is the distribution of LMNA and pLMNA around lipid droplets sexually dimorphic in either rats or human tissues in this study?

22. Related to comment #1, it would be helpful to show immunoblot data for all individual rat tissue samples in Figure 1B. 

33.Validation of primary cells is an important part of experimental design; it would be important to show that isolated AD-MSCs lack mature adipocyte markers and express expected AD-MSC marker(s). 

44. Figure 3E: anti-FLAG and beta-actin immunoblots are of poor quality (particularly the input blots). It is not clear to this reviewer why there is not an appreciable difference among any of the lanes when looking at the FLAG input blot in this panel. This makes the immunoprecipitation blots impossible to interpret – for example, it would appear that the FLAG antibody did not immunoprecipitate C-terminal or full-length tagged AMPK based on the top panels, but one cannot be sure due to the quality of the FLAG input blot. 

55. Figure 4C: It seems lonafarnib is poorly inhibiting farnesylation of prelaminA particularly in 7701 & MHCC 97-H cells. Therefore, it is not clear that inhibition of prelamin A farnesylation actually explains the observed changes in pAMPK or pACC1 (as opposed to off-target effects of lonafarnib). Dependence on prelamin A for the observed biochemical phenotype should be confirmed via additional controls (for example, perform the experiment in the setting of LMNA knockdown or knockout). This should be fairly trivial since the authors have LMNA KO 7701 cells available. 

66. Fig7C: In the text, it is indicated that pAMPK-T172 is increased in lonafarnib-treated cells; such data are shown in Fig 4C as referenced in comment #5, but there is no pAMPK blot shown in Figure 7. 

77. Direct evidence for the role of LMNA in NADH regeneration (as stated in the final paragraph of the introduction) are not presented. In the absence of such metabolite data, this claim should be removed.

Minor Comments:

11. Figure 2E and 3E: some protein molecular mass labels (kDa) are missing in these panels.

22. Figure 3E: labeled as 3F in the Figure Legend.

Round 2

Reviewer 1 Report

Although the authors did a great effort in answering the questions from this reviewer, I still see that LMNA splicing products are not clearly referred to and some information is misleading. For instance, which prelamin A do the authors refer to as "SC2, 661aa"? Or why lamin C is referred to as "SC1, 646aa"? Lamin C has 572 amino acids. It is hard to draw any conclusion on the data in this condition.

Author Response

Dear prof, 

Appreciate for your valuable suggestions, I have revised the manuscript according to these suggestions, detailed point to point reply, please see the attachment.

sincerely

Kong Lu

Reviewer 2 Report

The authors responded to this reviewer’s comments. However, some of those are still not enough to resolve this reviewer’s question.

1. About the pAMPKα-T172 IP-western in Figure 3D, there are many bands beside a band at 62 kD. If the authors say the band at 62 kD is specific to pAMPKα-T172, what are the bands other than it? The authors replied that they performed an exogenous reverse pull-down experiment. Please show the figure.

2. The authors did not answer to the following issue; in Figure 3E, the signal intensity of β-actin is different also in Input. This signal should be at least same in the four samples.

3. The authors answered the reason for elevating Lamin A/C after the treatment of Compound C as the treatment time was too short or higher dosage. This reviewer doesn’t understand the meaning of this reply. What is the mechanism of Lamin A/C elevation if the AMPK inhibitor treatment time is too short? And also what is the mechanism of Lamin A/C elevation if the dose of AMPK inhibitor is too high?

4. This reviewer could not find the graphical abstract. Where did the authors attach it?

Author Response

Dear Prof,

We admire your rigor and wisdom, and your suggestions have improved our manuscript considerably. Thank you very much for your suggestions, we have added experiments and answered questions in detail point by point, please see the attachment for details.

Sincerely

Kong lu

Reviewer 3 Report

In the revised manuscript entitled “Loss of mature lamin A/C triggers a shift in intracellular metabolic homeostasis via AMPKα activation,” the authors have improved their report of the molecular link between lamin phosphorylation and regulation of lipid metabolism showing that the active form of AMPK (AMPKα-T172) directly interacts with lamin A/C. Overall, this revised manuscript is significantly improved from the initial submission in response to referee comments.

Author Response

Dear prof,

Appreciate for your professional suggestion and your suggestions have improved our manuscript considerably. 

Thanks a lot

Sincerely

kong Lu

Round 3

Reviewer 1 Report

I went through the uniprot page indicated by the authors in their response. Unfortunately, the page reports several inaccurate or even wrong information. Prelamin C does not exist. Also, prelamin A/C does not exist. Only lamin A is produced as a precursor protein (of about 74 kDa, not 90 kDa)  called prelamin A. Lamin C is produced as mature protein of 572 aa. Further, progerin could be barely considered a lamin A isoform, it is an aberrant splicing product.

Speaking about LMNA products, this issue must be solved.

Author Response

Dear prof,

Thanks a lot. I really didn't understand the splicing product of LMNA correctly, and this time I do understand some experimental phenomena by reviewing the literature, please see the attachment for my detail reply.

Best regards

kong lu

Reviewer 2 Report

The authors appropriately replied to my questions.

The graphical abstract should be included in the main abstract or supplementary material for the reader to understand this article.

Author Response

Dear prof, 

Aprreicate for your suggestion to greatly improve our manuscript. About the graphical abstract, the editor replied that they will put the graphical abstract on the front page of the article webpage after our paper is accepted.

Best regards

kong lu